# Potential Pollution Sources from Agricultural Activities on Tropical Forested Floodplain Wetlands Revealed by Soil eDNA

**Maria Fernanda Adame** [1],*[ID] **and Ruth Reef** [2]

[1] Australian Rivers Institute, Griffith University, Nathan, QLD 4111, Australia
[2] School of Earth, Atmosphere and Environment, Monash University, Clayton, VIC 3800, Australia; ruth.reef@monash.edu
\* Correspondence: f.adame@griffith.edu.au

**Abstract:** Tropical floodplain wetlands are found in low-lying areas that are periodically inundated. During wet periods, these wetlands can receive large amounts of suspended and dissolved material from the catchment, including many potential pollutants. In this study, we use traditional isotope tracers ($\delta^{15}$N and $\delta^{13}$C) along with soil eDNA to investigate the sources of transported materials and potential contaminants in seven forested floodplain wetlands in tropical Australia. We hypothesised that eDNA and isotope tracers in the soil would reflect the land use of the catchment. Our goal was to test whether eDNA could be used as a potential tool to identify and monitor pollutants in floodplain wetlands. The sampling sites were located within catchments that have a mosaic of land types, from well-conserved rainforests to intensive agricultural land uses, such as grazing, sugar cane, wood production, and horticulture. The soil eDNA was comprised of a mix of plant species consistent with the land use of the catchments. Most of the eDNA pool was derived from native trees, accounting for 46.2 ± 6.5% of the total; while cultivated species associated with agricultural activities contributed to 1–24% of the total. From the cultivated species, highest contributions (>5%) were from *Sorghum* sp. used for grazing, banana (*Musa ornata*), melons (*Cucumis melo*), and *Pinus radiata* and *Juniperus* sp. grown for wood production. Interestingly, tropical wetlands on sites 15 km offshore had soil eDNA from agricultural activities of the mainland, highlighting the connectivity of these wetlands, probably during extensive floods. Overall, soil eDNA, more than isotopic tracers, showed promising results for tracing and monitoring potential pollutants in tropical floodplain wetlands that are highly connected and susceptible to environmental degradation.

**Keywords:** carbon; fertilisers; isotopes; *Melaleuca*; pesticides; tracers; weeds; water quality

## 1. Introduction

Forested floodplain wetlands are found in low-lying areas, where they are periodically inundated by the lateral overflow of rivers or lakes, or directly by rainfall and groundwater [1]. In tropical regions, these wetlands are highly productive and are habitat for many terrestrial and aquatic species [2,3]. During inundation periods, material from the catchment is transported and deposited within these wetlands [1,4]. As a result, floodplain forested wetlands can be considered sinks of nutrients and other pollutants, contributing to the improvement of water quality [5,6]. However, these wetlands can be threatened by the same material that they accumulate; excess nutrients and other pollutants can ultimately cause their degradation [7].

The amount and type of material deposited within a forested floodplain wetland depend on its location in the landscape, land-use of the catchment and sub-catchment, and local hydrology [8,9].

In catchments dominated by agricultural activities, fertilisers and pesticides can be rapidly mobilised during flooding episodes into waterways and eventually deposited in adjacent wetlands [4,10]. Due to the complexity of hydrology, the sources of these pollutants are usually difficult to trace. For instance, excess fertilisers and pesticides from cultivated land could end up hundreds or thousands of meters away from their sources [11], in a pathway dictated by hydrology and landscape configuration, not only by distance.

In the Great Barrier Reef catchments, intensive agricultural use has resulted in contamination of waterways by excessive nutrients and pesticides [12,13]. These pollutants can cause immediate or chronic effects on biota. For instance, pesticides can change the reproduction fitness of barramundi [14], and fertilisers can cause weed proliferation [15]. Currently, the assessment and monitoring of pollution in wetlands of the Great Barrier Reef catchments has been conducted through intensive temporal and spatial sampling [16,17] or the evaluation of the use of fertilisers and pesticides [12]. In these and other catchments with intense agricultural land-use, identifying the sources of contaminants at the species level could complement these efforts and be particularly useful to identify sources that are not necessarily close by. The identification of key species can be directly associated with a specific land-use activity, for example, horticulture of a particular fruit type, or pastures for cattle grazing. Information at the species level could also inform on the transport of invasive weeds, and to identify autochthonous versus the allochthonous contribution of carbon in the soil.

Various indicators have been used to trace pollutants as they are transported from their source across ecosystems. For instance, nitrogen isotopes ($\delta^{15}$N) have been successfully used to trace sewage outflows into seagrasses meadows [18]. Similarly, a combination of stable isotopes along with geochemical tracers (e.g., Zn, Pt and S) can distinguish soil derived from different agricultural activities, such as grazing, sugarcane, banana, or forestry [19]. However, tracers based on isotopes can have confounded and overlapping values among different plant species, which limits their use. Another indicator, glomalin, a protein produced by the mycorrhiza of trees, is a useful tracer of terrestrial material transport into the coastal zone but has limited applicability to distinguish the land use of its source [20,21]. More recently, eDNA has emerged as a promising tool to identify sources of organic material; it can be highly specific and can help identify at the species level plant material transported across different ecosystems [22].

In this study, we use eDNA and traditional isotopes ($\delta^{15}$N and $\delta^{13}$C) to trace the source of organic matter and potential pollutants of seven forested floodplain wetlands in tropical Australia. Five of these wetlands are located within catchments that have intensive agricultural activities, such as grazing, wood production, sugar cane, and horticulture, and are frequently flooded during the wet season. The other two sites are located on an offshore island (Hinchinbrook Island) which currently has no agricultural activities. We hypothesised that stable isotopes and eDNA in the wetland soil will reflect the land use of the catchment. Our goal was to test whether eDNA could be used as a tool for identifying sources and impacts of pollution without the need for large-scale and time-intensive monitoring.

## 2. Materials and Methods

### 2.1. Study Sites

Sampling was conducted within the Wet Tropics region, in the catchments of the Herbert and the Tully-Murray Rivers, northeast of Australia (Figure 1). These rivers discharge into the Coral Sea and the Great Barrier Reef. The climate is tropical with monthly mean temperatures ranging from 22 to 34 °C (1907–2018) and a mean total annual rainfall >2000 mm, mostly falling between December and May (1871–2018) [23]. The Herbert is drier than the Tully-Murray catchment with an annual rainfall of 2087 mm (1960–2017) compared to 2700 mm (1871–2017), respectively. The Herbert has a large catchment of 9846 km$^2$, of which 5.6% is wetlands (552 km$^2$); the Tully-Murray has a smaller catchment of 2792 km$^2$, of which 8.7% is wetlands (243 km$^2$) [24]. The rivers are characterised by relatively dry

periods with base groundwater flows during the winter months and periodic overbank floods in the summer that inundate these wetlands from 1 to 12 days at a time [25,26].

Both catchments are surrounded by mountains with tropical forests that are mostly well protected and conserved, and their coast is fringed by a network of protected wetlands that include the Hinchinbrook Channel and the World Heritage Hinchinbrook Island (Figure 1). The upper catchment of the Herbert River has extensive grazing, and the floodplains of both catchments have intensive agricultural land-use (Figure 1B). The land use for the Herbert Catchment is 56% grazing, 27% natural/minimal use, 8% sugarcane, 5% forestry, and 4% other land-uses [24]. For the Tully-Murray catchment, most of the land is natural/minimal land use (>65%), the remainder is 12% sugar cane, 12% urban and other land uses, 5% grazing, 4% bananas, 2% forestry, and 1% other crops [24] (Figure 2). There are sewage treatment plants in both catchments; in the Herbert catchment, treatment plants at Ingham and Lucinda discharge into Palm Creek and the Herbert River; in the Tully-Murray catchment, treated sewage is discharged into Banyan Creek in the town of Tully. The population is relatively low with 18,000 persons living within the Herbert catchment and about 3600 within the Tully-Murray catchment.

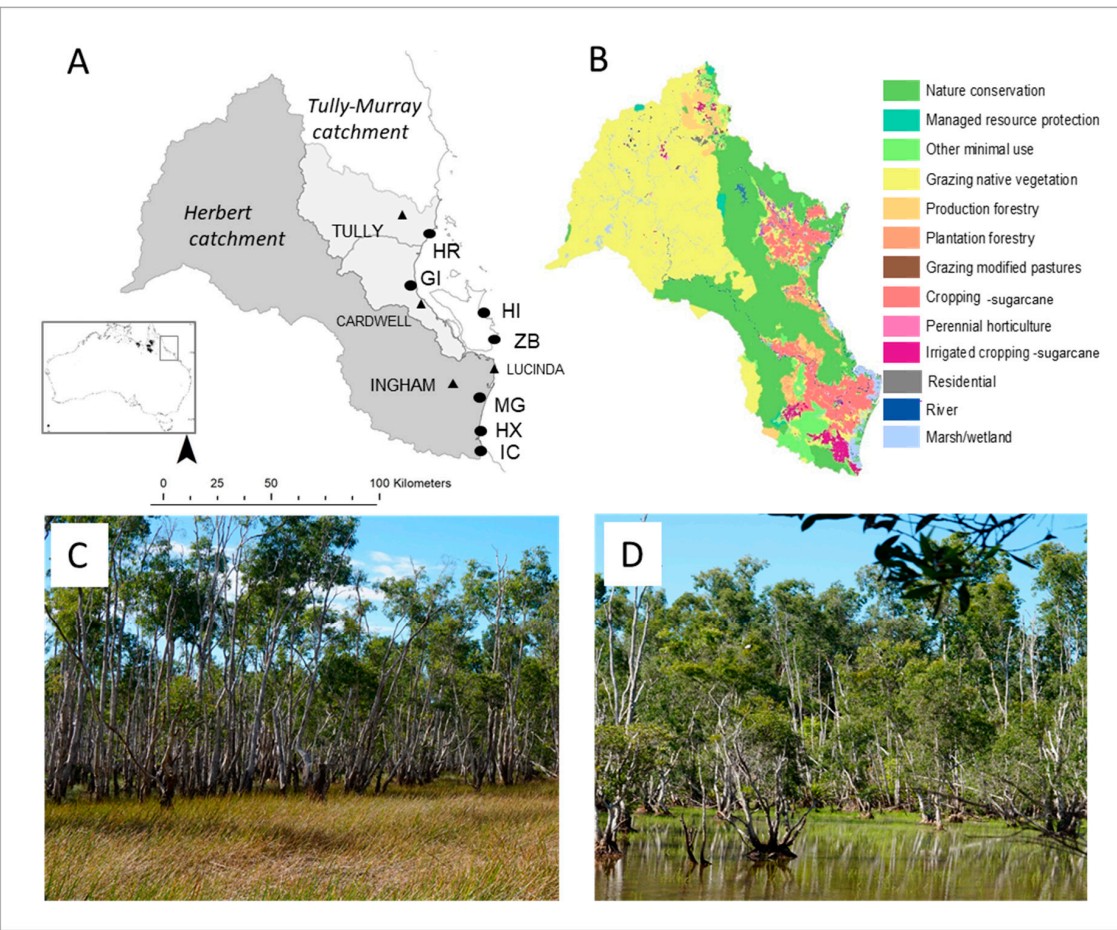

**Figure 1.** (**A**) Location of seven forested floodplain wetlands dominated by *Melaleuca* spp. in northeast Australia. Triangles represent main population centres: Tully (population of 2390), Cardwell (population of 1309), and Ingham (population of 4357); (**B**) Land uses for the Tully-Murray and Herbert River catchments. Data are from the Queensland Spatial Catalogue, Queensland Government, Australia [27]. (**C**) Girramay National Park, during the dry, and (**D**) wet season.

We sampled seven sites dominated by *Melaleuca* spp., commonly known as "tea tree swamps". This type of forested wetland occurs naturally in Australia and as an invasive ecosystem in

America [28,29]. One hectare of *Melaleuca* forest has the potential to accumulate annually 0.6 ton of carbon and 36 kg of nitrogen from autochthonous and allochthonous sources [6]. Stable isotopes and soil eDNA were sampled at each site (Table 1) in the dry seasons (May–October) of 2016 and 2017.

**Table 1.** Characteristics of the catchments, sub-catchments, vegetation and hydrology for sampling sites at seven forested floodplain wetlands dominated by *Melaleuca* spp. in northeast Australia.

| Site | Catchment | Sub-Catchment | Nearby Land Use and Impacts | Vegetation | Hydrology |
|------|-----------|---------------|-----------------------------|------------|-----------|
| (HR) Hull River National Park | Tully-Murray River | Hull River | Grazing Sugarcane Damaged during cyclone in 2011 | *M. viridiflora* Rainforest | Permanently flooded |
| (GI) Girramay National Park | | Whitfield-Dallachy Creek | Grazing Protected rainforest | *M. quinquenerva* Rainforest | Flooded by rain and overbank flows |
| (MG) Mungalla | Herbert River | Palm Creek | Grazing Treatment plant discharge | *M. quinquenerva* | Flooded by nearby lagoon |
| (HX) Halifax Wetlands National Park | | Trebonne Creek | Grazing Sugar cane Protected wetlands | *M. quinquenerva* Mangroves Marsh | Flooded by rain and tidal flows |
| (IC) Insulator Creek | | Insulator Creek | Sugar cane Protected rainforest | *M. quinquenerva* Mangroves Marsh | Flooded by rain and overbank flows |
| (HI) East Hinchinbrook Island | Hinchinbrook Island | East Hinchinbrook | Protected rainforest and beach | *M. quinquenerva* Rainforest | Flooded by rain and groundwater |
| (ZB) Zoe Bay | | Southeast Hinchinbrook | Protected wetlands and rainforests | *M. quinquenerva* Mangroves Marsh | Flooded by fringing tidal creek |

## 2.2. Soil Isotopes and Plant End-Members

Three soil samples (0–10 cm deep) were taken with a stainless-steel gouge auger of 4 cm diameter at the mainland sites (GA, Dormer Soil Samplers, NSW, Australia). Vegetation were hand-picked from the dominant groups found at each site, which included *M. quinquenerva*, *M. viridiflora*, ferns, sedges, and native grasses and trees. We chose green leaves from each vegetation type and the senescent litter of *Melaleuca* from the forest floor. These were considered the "end-members" of autochthonous material. We also collected nearby material from sugar cane and grazing pastures including the invasive grass *Hymenachne amplexicaulis*, as possible allochthonous sources of agricultural pollution. If the soil had significant input of material from allochthonous sources, we expected the soil isotope values to align towards these end-members. The soil and plants were kept refrigerated until transported back to the laboratory where they were oven-dried at 60 °C and ground. The bulk density of the soil samples was obtained from dry weight and volume. Soil samples were tested for the presence of inorganic carbon by adding 1M HCl to the soil. Samples were analysed for carbon (%C), nitrogen (%N), $\delta^{13}C$ and $\delta^{15}N$ with and without the addition of HCl in an elemental-analyser isotope ratio mass spectrometer (EA-IRMS, Serco System, Griffith University). The analytical standard deviation for the isotope measurements was 0.1‰ for $\delta^{13}C$ and below 0.2‰ for $\delta^{15}N$. More detailed results from the chemical analysis of soil samples can be found in Adame et al. [6].

## 2.3. eDNA Analyses

Ten surface soil replicates were collected from each sampling location except in Zoe Bay, where 20 replicates were taken in two plots which were 50 m apart from each other. The samples

were randomly collected within a 50 m plot established to account for soil heterogeneity at each site. The samples were frozen immediately following collection and kept at −20 °C for four weeks before processing. Genomic DNA was extracted from one mg soil with a DNeasy PowerSoil Kit (Qiagen, Hilden, Germany) following manufacturer protocols. Most samples yielded useable DNA for amplification with a minimum of 260/280 nm and a ratio of 1.8.

The PCR amplification and sequencing of the rbcL chloroplast gene, a widely used plant barcode [22], was performed by the Australian Genome Research Facility. The PCR amplicons were generated using the primers and conditions outlined in Supplementary Tables S1 and S2. Thermocycling was completed with an Applied Biosystem 384 Veriti and using Platinum SuperFi mastermix (Life Technologies, Australia) for the primary PCR. The first stage PCR was cleaned using magnetic beads and samples were visualised on 2% Sybr Egel (Thermo-Fisher Scientific, Waltham, MA, USA). A secondary PCR to index the amplicons was performed with TaKaRa PrimeStar Max DNA Polymerase (Clontech, Mountain View, CA, USA). The resulting amplicons were cleaned again using magnetic beads, quantified by fluorometry (Promega Quantifluor), and normalised. The equimolar pool was cleaned a final time using magnetic beads and measured with a High-Sensitivity D1000 Tape on an Agilent 2200 TapeStation. The pool was diluted to 5nM and molarity was confirmed again using a Qubit High Sensitivity dsDNA assay (Thermo-Fisher Scientific, Waltham, MA, USA). This was followed by sequencing on an Illumina MiSeq (San Diego, CA, USA) with a V2, 300 cycle kit (2 × 150 base pairs paired-end) and a 25% PhiX spike-in to improve nucleotide diversity.

The paired-ends reads were assembled by aligning the forward and reverse reads using PEAR (version 0.9.5) [30]. Primers were identified and trimmed. Trimmed sequences were processed using Quantitative Insights into Microbial Ecology (QIIME 1.8) [31] and USEARCH (version 8.0.16) [32,33]. Sequences were quality filtered and full-length duplicate sequences were removed and sorted by abundance. Singletons, or sequence that were present only once, were discarded as they were likely to be a sequencer error of the dataset. To obtain the number of reads in each Operational Taxonomic Unit (OTU), reads were mapped back to OTUs with a minimum identity of 97%. Taxonomy was assigned using the NCBI (National Center for Biotechnology Information) Blast (Basic Local Alignment Search Tool, https://blast.ncbi.nlm.nih.gov/Blast.cgi) database filtered to include species recorded from the bioregion in the Atlas of Living Australia (https://www.ala.org.au). Where <97% match was found with present local species, we blasted the nucleotide sequence against the entire NCBI database and a genus level identification was determined based on phylogenetic similarity to the species identified.

### 2.4. Data Analyses

We used a "mixing polygon" approach to identify possible contributions of the plant end-members to the total soil isotopic mix [34], which gives a similar result to mixing models [35]. We did not use a mixing model, as these would suggest that we included all possible sources to the soil mixture, which in this case, were mostly unknown at time of sampling [34]. We created a polygon with the plant end-member values as vertices of the polygon, and produced two plots ($\delta^{13}C$ vs $\delta^{15}N$, and $\delta^{13}C$ vs C:N) on which we overlaid the soil isotopic values. The contribution of each end-member to each site was estimated as the distance between soil sources and the end-members [36]. Data are presented as mean ± standard error.

## 3. Results

### 3.1. Soil Isotopes and End-Members

The site HX had the lowest soil organic C and N (1.0 ± 0.2% C, 0.06 ± 0.01% N, 0.84 ± 0.12 g cm$^3$), while HR had the highest soil C and N and the lowest bulk density (14.3 ± 1.4% C, 0.78 ± 0.04% N, 0.34 ± 0.05 g cm$^3$). Surface soil $\delta^{13}C$ values ranged from −30.3 to −22.4‰ with lowest values in HR and highest in HX, while $\delta^{15}N$ values were lowest in HR (−0.3 ± 0.6‰) and highest in GI (2.4 ± 0.4‰,

Table 2). The δ$^{15}$N values could not be measured in HX due to very low N concentrations. Data for the chemical analyses of soil for the offshore sites (ZB and HI) were not available.

**Table 2.** Surface (0–10 cm) soil characteristics (BD = bulk density, N = nitrogen, C = organic carbon, C:N = atomic ratio) of five forested floodplain wetlands, n.d. = not detected. More detailed results from the chemical analysis of soil samples can be found in Adame et al. [6]. The full names and location of the sites can be found in Table 1 and Figure 1.

| Sites | BD (g cm$^{-3}$) | N (%) | C (%) | C:N | δ$^{13}$C (‰) | δ$^{15}$N (‰) |
|---|---|---|---|---|---|---|
| HR | 0.34 ± 0.05 | 0.78 ± 0.04 | 14.3 ± 1.4 | 22.4 ± 2.2 | −30.3 ± 0.6 | −0.3 ± 0.6 |
| GI | 0.59 ± 0.02 | 0.34 ± 0.02 | 4.1 ± 0.4 | 23.8 ± 0.4 | −25.7 ± 1.0 | 2.4 ± 0.4 |
| MG | 0.61 ± 0.07 | 0.44 ± 0.15 | 7.6 ± 2.8 | 21.7 ± 0.7 | −25.5 ± 2.1 | 1.6 ± 0.9 |
| HX | 0.84 ± 0.12 | 0.06 ± 0.01 | 1.0 ± 0.2 | 20.6 ± 0.3 | −22.4 ± 1.1 | n.d. |
| IC | 0.65 ± 0.10 | 0.54 ± 0.14 | 7.8 ± 2.2 | 18.7 ± 1.2 | −25.5 ± 0.8 | 2.3 ± 0.4 |

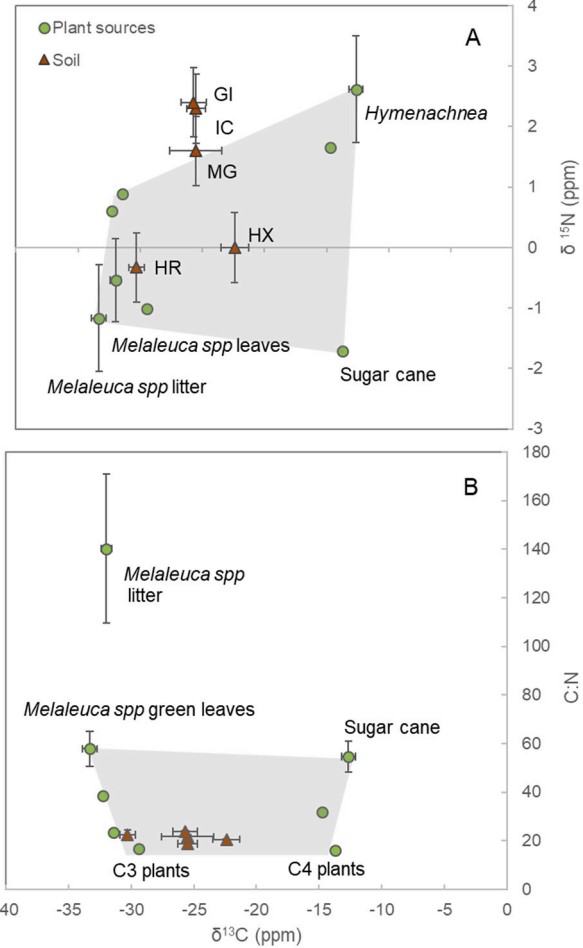

**Figure 2.** Surface (0–10 cm) soil (triangles) values for (**A**) δ $^{13}$C versus δ $^{15}$N; and (**B**) δ $^{13}$C versus C:N of five forested floodplain wetlands, and potential end-members (circles) which included green and senescent leaves of *Melaleuca* trees, sugar cane, ferns, *Hymenachne amplexicaulis*, and other grass and shrubs picked from the sites. Values are shown as mean ± standard error of triplicate samples. The grey area comprises the isotopic "mixing polygon" (see Methods). The full names and location of the sampling sites can be found in Table 1 and Figure 1.

The δ$^{13}$C values separated the end-members into two groups corresponding to plants with C$_3$ (−34.2 to −29.4‰) or C$_4$ (−14.7 to −12.7‰) metabolism, the latter encompassing most grasses, such as

sugarcane (Figure 2). The $\delta^{15}$N isotopes further isolated the sugarcane leaves, which had the lowest $\delta^{15}$N value with −1.7‰. Finally, the C:N of end-members separated *Melaleuca* leaves into green and senescent (Figure 2B).

When overlying values from the soil to the isotope polygon, it was challenging to assign sources to the mix unequivocally. The soil of most sites was either close to the centre (HX) or outside the polygon (GI, IC and MG), suggesting that some sources were not captured in our field collection. Senescent *Melaleuca* litter was far outside the end-member polygon and did not contribute substantially to the soil isotopic mix. The only exception was HR, the soil of which had an isotopic composition very similar to green leaves of *Melaleuca* trees.

*3.2. eDNA*

Most of the eDNA pool was of native trees, accounting for 46.2 ± 6.5% of the total, followed by microalgae (13.8 ± 3.2%), cultivated species (13.4 ± 3.1%), macrophytes (3.9 ± 2.6%), native grass and shrubs (3.9 ± 2.6%), and non-native weeds (3.9 ± 2.6%, Table 3). The cultivated species from agriculture had a contribution of 1–24% to the total eDNA pool of all wetlands.

**Table 3.** Percentage (%) contribution of native terrestrial and aquatic plants, versus non-native and cultivated species to the total eDNA pool of the soil of seven forested floodplain wetlands. The full names and location of the sampling sites can be found in Table 1 and Figure 1.

| Site | Native Trees | Grass/Shrubs | Macrophyte | Microalgae | Non-Native | Cultivated |
|------|------|------|------|------|------|------|
| HR | 21.5 | 0.03 | 0.01 | 0.04 | 0.01 | 16.0 |
| GI | 50.7 | 4.9 | 0.1 | 20.0 | 0.1 | 20.7 |
| MG | 57.8 | 0.8 | 17.2 | 18.9 | 0.9 | 1.4 |
| HX | 57.1 | 0.2 | 9.6 | 22.1 | n.d. | 6.8 |
| IC | 30.6 | 35.7 | 0.02 | 15.1 | 0.01 | 17.4 |
| HI | 69.6 | 1.9 | 0.03 | 4.0 | 0.3 | 7.5 |
| ZB | 35.8 | 0.1 | 0.6 | 16.7 | 0.5 | 24.2 |

The differences among sites reflected some of their hydrological and ecological characteristics. For instance, MG had the highest contribution of freshwater macrophytes (17.2%) consistent with this site being next to a deep lagoon. The site IC had the highest contribution of native grass and shrubs (35.7%), and IC, GI and HR had the highest contribution of cultivated species (>15%). The sites in Hinchinbrook Island (ZB and HI) had similar or higher contributions from agricultural activities as some sites in the mainland, even though they were located >15 km offshore (Table 3).

The species identified in the eDNA pool were mostly consistent with the characteristics of the sites (Figure 3). *M. quinquenervia* was found at all the sites. Other native tropical trees with high contributions were *Ormosia ormondii* ("yellow bean"), *Elattostachys xylocarpa* ("white tamarind"), *Syzygium hemilamprum* ("cassowary gum"), *Dillenia alata* ("red beech"), *Hibiscus tiliaceus* ("coastal hibiscus") and *Diospyros* sp. We identified three invasive weeds within our sites: *Tephrosia* sp. at HI, *Romulea osbscura* at ZB, HI and MG and *Urena lobata* at ZB, although their contribution to the total eDNA pool was low, with less than 1 % of the total.

For macrophytes, the highest contribution to the eDNA pool was from freshwater *Glinus oppositifolius* ("sweetjuice") and *Philydrum lanuginosum* ("frogsmouth"). At HX, a wetland near marshes and mangroves, the contribution of *Hydrilla verticillata*, a macrophyte with high salinity tolerance, was substantial (8.0%). Freshwater microalgae constituted a large proportion of the eDNA pool at HX, GI and ZB, with highest contributions for *Lepocinclis tripteris* (7.7 ± 7.5%) at HX, *Euglena mutabilis* (9.5 ± 9.5%) at IC, *Trachelomonas volvocina* at MG (9.2 ± 5.6%), and *Coccomyxa* cf. *olivacea* at GI (16 ± 10%) and at ZB (11 ± 4.8%). The full list of species can be found in the Supplementary Data file.

We identified 16 species associated with agricultural activities, three species of pasture, *Sorghum* sp., *Trifolium repens* ("white clover") and *Vicia* sp. (vetch), two from forestry: *Pinus radiata* ("radiated pine")

and *Juniperus* sp., and other species for horticulture and other crops. We identified banana (*Musa ornata*), melon (*Cucumis melo*), sugar cane (*Saccharum officinarum*), kiwi (*Actinidia* sp.), cappers (*Capparis* sp.), pineapple (*Ananas comosus* var. *bracteatus*), macadamia (*Macadamia ternifolia*), carrots (*Daucus carota*), mango (*Mangifera indica*), and lemon (*Citrus limon*). The highest contribution for cultivated species (>5%) was from *Sorghum* sp. at IC and GI, *Juniperus* sp. at IC, and melons in HR and GI. In the offshore site of ZB, *Sorghum* sp. contributed 15.2 ± 5.0% to its eDNA pool. Overall, 13.6 ± 3.0% of the eDNA pool was associated with agricultural activities.

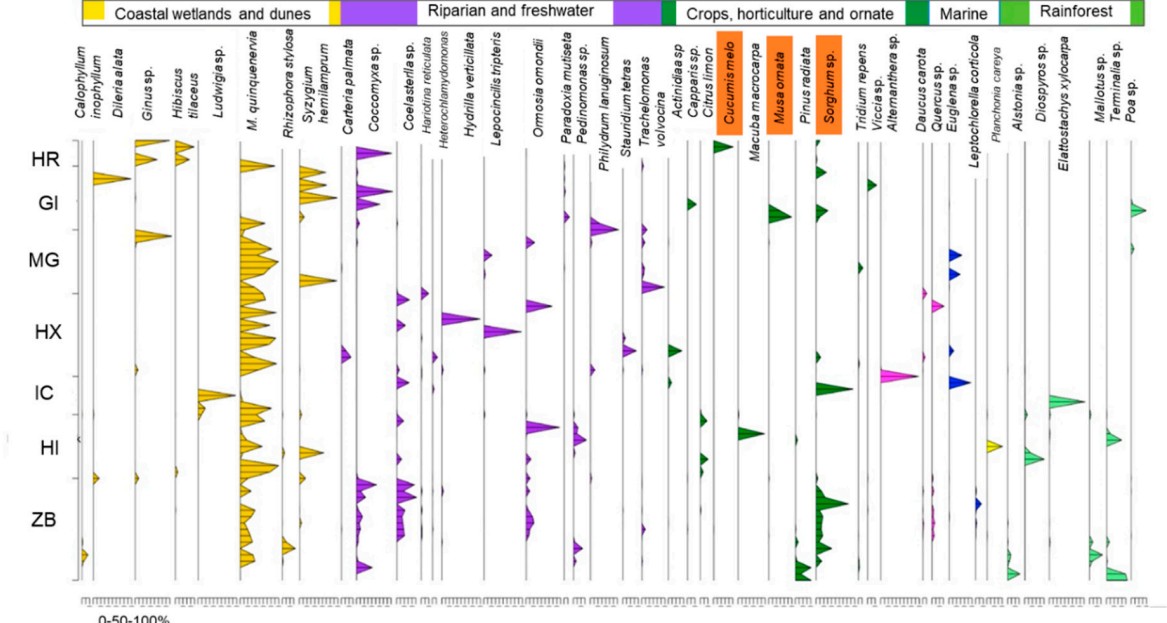

**Figure 3.** Relative abundance (*x*-axis, % of total) of Operational Taxonomic Unit (OTU) of surface soils of seven tropical forested floodplain wetlands (*y*-axis). The tree species *Melaleuca quinquenervia* was the most abundant OTU. Key species associated with agricultural activities within the catchment are highlighted: melons (*Cucumis melo*), banana (*Musa ornata*), and *Sorghum* sp. Only species with a minimum total DNA abundance of 10% are included. The full names and location of the sampling sites can be found at Table 1 and Figure 1.

## 4. Discussion

We identified plant eDNA from agricultural activities in seven floodplain forested wetlands consistent with the land-uses of the catchment and suggested potential sources of pollution. The three most common plant species associated with agricultural activities were *Sorghum* sp. used for grazing, banana (*Musa ornata*) and melons (*Cucumis melo*) for horticulture, and radiated pine (*Pinus radiata*) and juniper (*Juniperus* sp.) for wood production. The eDNA of sorghum was found as far as Hinchinbrook Island, a site 15 km offshore.

The eDNA results from this study support the idea of interconnectedness between floodplain wetlands and their wider river catchment. Most of the vegetation types identified in the soils were consistent with the land-use of their wider catchment. However, not all the land-uses were captured with this method. For instance, sugarcane is a dominant crop (12%) of the Tully-Murray catchment, and comprises 8% of land use in the Herbert catchment [24]. In IC, sugarcane fields are less than 100 m away from our sampling site, but eDNA for sugarcane was only found in small quantities (<1%). The low abundance of this grass species, despite their ubiquitous presence in the region, suggests that eDNA from sugarcane might be difficult to extract or is less stable in the soil. Another constraint of the eDNA analyses is that the identification of species was limited to the availability of genetic libraries.

For instance, *M. viridiflora* was not in the library, and thus, was not detected at HR, a site dominated by this species.

The eDNA of sorghum appears to be useful to trace the impacts of grazing in offshore areas. Sorghum was found at ZB in substantial quantities (15% of the total soil pool), higher than some mainland sites adjacent to pastures (e.g., HX and MG). The site at ZB is over 15 km from the mouth of the Herbert River, but during major floods, it is in the path of a plume that flows northwest and passes close to the site [37]. This plume could transport sediment from large grazing areas in the upper Herbert catchment (Figure 1B) all the way to the east coast of Hinchinbrook Island. Similar results have been found within this coastal area, where terrestrial material has been detected 60 km offshore [20]. In contrasts, floodplain forests at HX had no detectable contribution of agricultural activities, even though the site is adjacent to a grazing pasture. A plausible explanation is that this site is mostly influenced by tidal, not surface, runoff. Additional explanations for these seemingly unexpected connectivity pathways are aeolian and human transportation of plant material. Overall, these findings support the idea that connectivity of floodplain forests is not always directed by distance, but also by other factors such as hydrology, geomorphology, configuration of the sub-catchment, and human activities.

Besides identifying potential sources of pollutants, eDNA was useful to separate autochthonous from allochthonous sources of soil carbon. For instance, at MG and HX, a large portion of the eDNA pool was from aquatic macrophytes, which are allochthonous sources. This information is useful when establishing the value for carbon sequestration of wetlands, which requires considering how much carbon is produced and how much is transported from elsewhere [38]. Finally, eDNA was useful as a monitoring tool to identify potential invasive species. Weed invasion is a severe problem in these wetlands, with the Herbert River having recorded 351 introduced and possibly invasive plant species [24]. We identified two potential invasive species with significant contributions to the eDNA pool: *Tephrosia* sp. at HI, and *Romulea obscura* at MG, the latter only recorded in Western Australia until now. This information could serve as an early warning for potential new invasive species or could be used as an indicator of success/failure of restoration practices, such as the removal of invasive plants [15].

The analyses of isotopes gave limited insights into sources of the soil eDNA pool, mainly due to overlapping values of end-members and multiple potential sources that were not sampled. The $\delta^{13}$C isotope values were useful to identify plants according to their metabolism ($C_3$ or $C_4$), and C:N helped separate green from senescent leaves. However, most sites fell either at the centre or outside the mixing polygon, limiting our capacity to determine sources unambiguously [34]. In other studies, stable isotopes have been successful to determine sources of carbon in the soil, but only when there was a clear alignment with the soil and the values of the dominant species [39]. In our case, it was clear that the dominant species (*Melaleuca* spp.) was not the dominant contributor of organic matter to the soil [6]. Interestingly, we were missing one source with high $\delta^{15}$N values, suggesting N derived from animals or sewage. Because untreated sewage, which is highly regulated in the country, is unlikely to reach these wetlands, the most likely source for these high $\delta^{15}$N values is pigs, an invasive species with large populations permanently settled in most wetlands of the region [15]. These results support the idea that isotopes cannot be used alone to identify sources when they are numerous and have overlapping values [34].

## 5. Conclusions

Our analyses of soil eDNA have shown that forested floodplain wetlands in tropical Australia receive various potential sources of pollution from agricultural activities within their catchment. Our results also confirm the high interconnectivity of these forests, and the potential to receive pollutants from sources that are tens of kilometers away, especially during periods of extensive flooding. Tropical floodplain wetlands are highly interconnected ecosystems capable of sequestering carbon, improving water quality and providing habitat for a range of unique species. However,

this interconnectedness makes them vulnerable to pollution from their catchments. Soil eDNA is a promising tool to track inputs and to monitor management actions aiming at reducing pollution in these and similar connected ecosystems.

**Supplementary Materials:** The following are available online at http://www.mdpi.com/1999-4907/11/8/892/s1.

**Author Contributions:** M.F.A. conceptualisation, formal analyses, funding acquisition, project administration, writing original draft, R.R. conceptualisation, formal analyses, methodology, writing review and editing. All authors have read and agreed to the published version of the manuscript.

**Funding:** This research was funded by the Queensland Government, Australia.

**Acknowledgments:** We want to acknowledge the Traditional Owners on the land on which this study was conducted, especially the Nywaigi, Djiru, Girramay, and Gulnay people. We thank the Queensland Government through the Advance Queensland Fellowship granted to MFA, the Great Barrier Reef Water Quality Program for their financial support, and the Australian Government's National Environmental Science Program, Tropical Water Quality Hub (Project 3.3.2).

**Conflicts of Interest:** The authors declare no conflict of interest.

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
