# Peer review of "Potential Pollution Sources from Agricultural Activities on Tropical Forested Floodplain Wetlands Revealed by Soil eDNA"

_forests, doi:10.3390/f11080892_

Round 1
Reviewer 1 Report
In the abstract, it would be necessary to clearly specify the purpose and objectives of the study.
In the introduction, it would be necessary to show which are the similar studies that approached the subject of pollutants (a chronological approach), which are the methods used in these studies compared to those used by the authors, which are the novelty elements brought from current study.
Also, due to the fact that five forests are located in areas with intense agricultural activities, it is necessary to insist on the detailed presentation of these types of activities (it was done succinctly in the methodology), in accordance with the specifics of each area, highlighting what is the impact that pollutants have in relation to these activities. In the same way, the role of the local population must be explained, what type of communities are in terms of demographic size, what are the functions of the settlements in the studied area. Perhaps, it would have been useful to locate human settlements and cultivated areas in fig. 1, for the reader to have an overview.
Even if the discussions are relevant and interesting, the authors must formulate a chapter of final conclusions.
Author Response
Thank you very much for the suggested revisions, see below a detailed response to each of the comments
In the abstract, it would be necessary to clearly specify the purpose and objectives of the study.
We have included in the abstract the hypothesis and goal of the study.
L17: “We hypothesised that eDNA on the soil would reflect the land use of the catchment. Our goal was to test whether eDNA could be used as a potential tool to identify and monitor pollutants in floodplain wetlands”
In the introduction, it would be necessary to show which are the similar studies that approached the subject of pollutants (a chronological approach), which are the methods used in these studies compared to those used by the authors, which are the novelty elements brought from current study.
We have included a new paragraph in the Introduction which explain the current approach to monitor pollutants and how it differs from our proposed method.
L55: “…..Currently, the assessment and monitoring of pollution in wetlands of the Great Barrier Reef catchments has been conducted through intensive temporal and spatial sampling [16,17] or the evaluation of the trends of use of fertilisers and pesticides [12]. In these and other catchments with intense agricultural land-use, identifying the sources of contaminants at the species level could complement these efforts and be particularly useful to identify sources that are not necessarily close by. The identification of key species can be directly associated with a specific land-use activity, e.g. horticulture of a particular fruit type, or pastures for cattle grazing. Information at the species level could also inform on the transport of invasive weeds, and to identify autochthonous versus the allochthonous contribution of carbon in the soil.”
Also, due to the fact that five forests are located in areas with intense agricultural activities, it is necessary to insist on the detailed presentation of these types of activities (it was done succinctly in the methodology), in accordance with the specifics of each area, highlighting what is the impact that pollutants have in relation to these activities. In the same way, the role of the local population must be explained, what type of communities are in terms of demographic size, what are the functions of the settlements in the studied area. Perhaps, it would have been useful to locate human settlements and cultivated areas in fig. 1, for the reader to have an overview.
We have included a new Figure (Fig. 2) where we have mapped the most important land uses for each catchment, including urban and residential areas. We have also included the main local population centres and their demographic size within Figure 1.
Figure 1. …”Triangles represent main population centres: Tully (population of 2,390), Cardwell (population of 1,309), and Ingham (population of 4,357).”
Finally, we have highlighted in the Introduction the potential impacts of these land uses on wetlands
L51: “In the Great Barrier Reef catchments, intensive agricultural use has resulted in contamination of waterways by excessive nutrients and pesticides [12,13]. These pollutants can cause immediate or chronic effects on biota. For instance, pesticides can change the reproduction fitness of barramundi [14], and fertilisers can cause weed proliferation [15].”
Even if the discussions are relevant and interesting, the authors must formulate a chapter of final conclusions.
We have included a Conclusion section at the end of the Discussion:
L 348. Conclusion:
“The analyses of soil eDNA have shown that these tropical forested wetlands receive various potential sources of pollution from agricultural activities within their catchment. Our results also confirm the high interconnectivity of these forests, and the potential to receive pollutants from sources that are tens of kilometres away, especially during periods of extensive flooding. Tropical floodplain wetlands are highly interconnected ecosystems capable of sequestering carbon, improving water quality and providing habitat for a range of unique species. However, this interconnectedness makes them vulnerable to pollution from their catchments. Soil eDNA is a promising tool to track inputs and to monitor management actions aiming at reducing pollution in these and similar connected ecosystems.”
Reviewer 2 Report
The authors present a very interesting study with a crucial topic for wetlands forests. They showed the utility of eDNA to track pollutants from different land-uses. However, or the authors do not explain their methodology or their eDNA analysis is a lack of robustness.
In the methods, they said that used fastq files to blast against NCBI database (lines 177-178). There are several steps that need to be done between fastq files until the blast with databases. The authors need to remove adapter and primers, filter reads by quality, filter chimeric sequences, cluster the sequence in Operational Taxonomic Units, or Average Sequence Variants (or both), remove singletons. After all these steps they can check with the database, such as NCBI, but they should get the similarity threshold used, it is good also to include it in a supplementary table.
DNA metabarcoding produces many biased reads, the clean steps are fundamental in any studies. Just check fastq files against NCBI, a database that is full of erroneous sequences and misidentification make all results and discussion potentially wrong. Without the filter quality, error check and similarity threshold clear in the methods I can not trust in any of the present results. I suggest some basics (among so many others) papers that discuss the prospects and limitations of eDNA studies.
Creer, S., Deiner, K., Frey, S., Porazinska, D., Taberlet, P., Thomas, W.K., Potter, C. and Bik, H.M., 2016. The ecologist's field guide to sequence‐based identification of biodiversity. Methods in Ecology and Evolution, 7(9), pp.1008-1018.
Zinger, L., Bonin, A., Alsos, I.G., Bálint, M., Bik, H., Boyer, F., Chariton, A.A., Creer, S., Coissac, E., Deagle, B.E. and De Barba, M., 2019. DNA metabarcoding—Need for robust experimental designs to draw sound ecological conclusions. Molecular ecology, 28(8), pp.1857-1862.
Prodan, A., Tremaroli, V., Brolin, H., Zwinderman, A.H., Nieuwdorp, M. and Levin, E., 2020. Comparing bioinformatic pipelines for microbial 16S rRNA amplicon sequencing. Plos one, 15(1), p.e0227434.
Other minor comments:
Lines 110-140: The description of the areas could be presented in a table.
Line 143: How the triplicates were collected, any specific design?
Line 160: How the ten replicates were collected, the spatial design should be described. Why Zoe Bay has 20 replicates?
Figure 3 is impossible to read the text in the figure.
Author Response
Thank you very much for your comments and suggestions, we have responded to each one of them below.
Reviewer 2.
The authors present a very interesting study with a crucial topic for wetlands forests. They showed the utility of eDNA to track pollutants from different land-uses. However, or the authors do not explain their methodology or their eDNA analysis is a lack of robustness.
In the methods, they said that used fastq files to blast against NCBI database (lines 177-178). There are several steps that need to be done between fastq files until the blast with databases. The authors need to remove adapter and primers, filter reads by quality, filter chimeric sequences, cluster the sequence in Operational Taxonomic Units, or Average Sequence Variants (or both), remove singletons. After all these steps they can check with the database, such as NCBI, but they should get the similarity threshold used, it is good also to include it in a supplementary table.
DNA metabarcoding produces many biased reads, the clean steps are fundamental in any studies. Just check fastq files against NCBI, a database that is full of erroneous sequences and misidentification make all results and discussion potentially wrong. Without the filter quality, error check and similarity threshold clear in the methods I can not trust in any of the present results. I suggest some basics (among so many others) papers that discuss the prospects and limitations of eDNA studies.
Creer, S., Deiner, K., Frey, S., Porazinska, D., Taberlet, P., Thomas, W.K., Potter, C. and Bik, H.M., 2016. The ecologist's field guide to sequence‐based identification of biodiversity. Methods in Ecology and Evolution, 7(9), pp.1008-1018.
Zinger, L., Bonin, A., Alsos, I.G., Bálint, M., Bik, H., Boyer, F., Chariton, A.A., Creer, S., Coissac, E., Deagle, B.E. and De Barba, M., 2019. DNA metabarcoding—Need for robust experimental designs to draw sound ecological conclusions. Molecular ecology, 28(8), pp.1857-1862.
Prodan, A., Tremaroli, V., Brolin, H., Zwinderman, A.H., Nieuwdorp, M. and Levin, E., 2020. Comparing bioinformatic pipelines for microbial 16S rRNA amplicon sequencing. Plos one, 15(1), p.e0227434.
Thank you for the suggestion, we acknowledge that our Methods were not clearly explained in the first version. We have included all the steps that were taken between fastq files until the blast with databases and all the filter qualities that were done to our database to ensure that the results are reliable and robust.
L166: “The PCR amplification and sequencing of the rbcL chloroplast gene, a widely used plant barcode [22], was performed by the Australian Genome Research Facility. The PCR amplicons were generated using the primers and conditions outlined in Supplementary Tables 1 and 2. Thermocycling was completed with an Applied Biosystem 384 Veriti and using Platinum SuperFi mastermix (Life Technologies, Australia) for the primary PCR. The first stage PCR was cleaned using magnetic beads, and samples were visualised on 2% Sybr Egel (Thermo-Fisher Scientific, MA, USA). A secondary PCR to index the amplicons was performed with TaKaRa PrimeStar Max DNA Polymerase (Clontech, CA, USA). The resulting amplicons were cleaned again using magnetic beads, quantified by fluorometry (Promega Quantifluor) and normalised. The eqimolar pool was cleaned a final time using magnetic beads and measured with a High-Sensitivity D1000 Tape on an Agilent 2200 TapeStation. The pool was diluted to 5nM and molarity was confirmed again using a Qubit High Sensitivity dsDNA assay (Thermo-Fisher Scientific, MA, USA). This was followed by sequencing on an Illumina MiSeq (San Diego, CA, USA) with a V2, 300 cycle kit (2 x 150 base pairs paired-end) and a 25% PhiX spike-in to improve nucleotide diversity.
The paired-ends reads were assembled by aligning the forward and reverse reads using PEAR (version 0.9.5)[30]. Primers were identified and trimmed. Trimmed sequences were processed using Quantitative Insights into Microbial Ecology (QIIME 1.8)[31] and USEARCH (version 8.0.16) [32,33]. Sequences were quality filtered and full-length duplicate sequences were removed and sorted by abundance. Singletons or unique reads in the data set were discarded. To obtain number of reads in each OTU, reads were mapped back to OTUs with a minimum identity of 97%. Taxonomy was assigned using NCBI Blast database filtered to include species recorded from the bioregion in the Atlas of Living Australia (https://www.ala.org.au). Where <97% match was found with present local species, we blasted the nucleotide sequence against the entire NCBI database and a genus level identification was determined based on phylogenetic similarity to species identified by the NCBI database..”
We have also included Supplementary information on the primers used and the thermocycling conditions (Table S1 and S2). Finally, we have included as supplementary data all the species identified in our analyses.
Other minor comments:
Lines 110-140: The description of the areas could be presented in a table.
We have made a new Table 1, where we describe the main characteristics of each site
Line 143: How the triplicates were collected, any specific design?
Line 160: How the ten replicates were collected, the spatial design should be described. Why Zoe Bay has 20 replicates?
We have explained the design of the sample collection
L160: “Ten surface soil replicates were collected from each sampling location except Zoe Bay, where 20 replicates were taken as two sites were sampled. The samples were randomly collected within a 50 m plot established to account for soil heterogeneity at each site.”
Figure 3 is impossible to read the text in the figure.
We have increased the font of Figure 3 (Fig. 4 in this new version) to make the text legible
Round 2
Reviewer 1 Report
The authors have made consistent improvements to the article, according to the previous request. The restructuring of the article led to clearly presented results and relevant conclusions.
Author Response
Thank you very much for taking the time to review this manuscript and for your helpful input.
Reviewer 2 Report
The author explain better their methods and I am satisfying with the new version. Couple small comments are:
Table 1 is not a table,or it is a box or should be formatted as table 2 and 3.
Line 166: "Singletons or unique reads in the data set were discarded." What is the different of singletons and unique reads? Do you refer to OTUs when say singleton? It needs be clarified.
Line 251: OTU instead OUT.
A more general comments the isotopes are very little discussed, I suggest or amplify a little a discussion of the potential and limitation of the use of isotopes or move the results for a supplementary material.
Author Response
Thank you very much for your time in reviewing this manuscript. We have revised our manuscript as described below.
Table 1 is not a table,or it is a box or should be formatted as table 2 and 3.
We have changed the formatting of Table 1 to clarify that it is a Table
Line 166: "Singletons or unique reads in the data set were discarded." What is the different of singletons and unique reads? Do you refer to OTUs when say singleton? It needs be clarified.
We have clarified that singletons are not OTUs but a read with a sequence that is present exactly once, i.e. is unique among the reads – which means they are a sequencer error, not a real sequence.
L185: “. Singletons, or sequence that were present only once were discarded, as they were likely to be a sequencer error of the dataset”
Line 251: OTU instead OUT.
We have changed this typo, that’s for pointing it out.
A more general comments the isotopes are very little discussed, I suggest or amplify a little a discussion of the potential and limitation of the use of isotopes or move the results for a supplementary material.
We have expanded our Discussion on the isotope results:
L340: “The analyses of isotopes gave limited insights into sources of the soil eDNA pool, mainly due to overlapping values of end-members and multiple potential sources that were not sampled. The δ13C isotope values were useful to identify plants according to their metabolism (C3 or C4), and C:N helped separate green from senescent leaves. However, most sites fell either at the centre or outside the mixing polygon, limiting our capacity to determine sources unambiguously [34]. In other studies, stable isotopes have been successful to determine sources of carbon in the soil, but only when there was a clear alignment with the soil and the values of the dominant species [39]. In our case, it was clear that the dominant species (Melaleuca spp) was not the dominant contributor of organic matter to the soil [6]. Interestingly, we were missing one source with high δ15N values, suggesting N derived from animals or sewage. Because untreated sewage, which is highly regulated in the country, is unlikely to reach these wetlands, the most likely source for these high δ15N values is pigs, an invasive species with large populations permanently settled in most wetlands of the region [15]. These results support the idea that isotopes cannot be used alone to identify sources when they are numerous and have overlapping values [34].”